# Stereospecificity of Ginsenoside AD-1 and AD-2 Showed Anticancer Activity via Inducing Mitochondrial Dysfunction and Reactive Oxygen Species Mediate Cell Apoptosis

**DOI:** 10.3390/molecules28186698

**Published:** 2023-09-19

**Authors:** Xude Wang, Meng Ding, Hong Zhao, Mengru Zhou, Xuan Lu, Yuanyuan Sun, Qinggao Zhang, Yuqing Zhao, Ruoyu Wang

**Affiliations:** 1Department of Oncology, The Affiliated Zhongshan Hospital of Dalian University, Dalian 116001, China; xude-wang@hotmail.com; 2Chronic Disease Research Center, Medical College, Dalian University, Dalian 116622, China; 15199366229@163.com; 3College of Chemistry and Chemical Engineering, Cangzhou Normal University, Cangzhou 061000, China; dm893332977@126.com; 4China College of Life and Health, Dalian University, Dalian 116622, China; zhaohong@dlu.edu.cn (H.Z.); luxuan_232@163.com (X.L.); 5School of Functional Food and Wine, Shenyang Pharmaceutical University, Shenyang 110016, China; sunyuanyuan921104@163.com; 6Key Laboratory of Natural Medicines of the Changbai Mountain, Ministry of Education, Yanbian University, Yanji 133002, China; zhangqinggao@dlu.edu.cn

**Keywords:** AD-1 and AD-2, isomers, anti-cancer, apoptosis, ROS

## Abstract

In this paper, the anti-cancer activity and molecular mechanisms of the isomers of AD-1 and AD-2 (20(*R*)-AD-1, 20(*R*)-AD-2, 20(*S*)-AD-1 and 20(*S*)-AD-2) were investigated. The results indicated that all of the four compounds obviously suppressed the viability of various cancer cells, and the anti-cancer activity of 20(*R*)-AD-1 and 20(*R*)-AD-2 was significantly better than 20(*S*)-AD-1 and 20(*S*)-AD-2, especially for gastric cancer cells (BGC-803). Then, the differences in the anti-cancer mechanisms of the isomers were investigated. The data showed that 20(*R*)-AD-1 and 20(*R*)-AD-2 induced apoptosis and decreased MMP, up-regulated the expression of cytochrome C in cytosol, transferred Bax to the mitochondria, suppressed oxidative phosphorylation and glycolysis and stimulated reactive oxygen species (ROS) production. Apoptosis can be attenuated by the reactive oxygen species scavenger N-acetylcysteine. However, 20(*S*)-AD-1 and 20(*S*)-AD-2 barely exhibited the same results. The results indicated that 20(*R*)-AD-1 and 20(*R*)-AD-2 suppressed cellular energy metabolism and caused apoptosis through the mitochondrial pathway, which ROS generation was probably involved in. Above all, the data support the development of 20(*R*)-AD-1 and 20(*R*)-AD-2 as potential agents for human gastric carcinoma therapy.

## 1. Introduction

Cancer poses a huge threat to public health with a highly increasing incidence and mortality all over the world. The malignant properties of cancer, especially metastasis, are beyond the potential of surgical removal or local ablation via radiotherapy. Chemotherapy, which is a systemic drug-based method, remains irreplaceable because of the limitations of the two approaches [1].

ROS are constantly produced during cellular metabolism, specifically through mitochondrial respiration, and elicit various physiological effects. Recent investigations propose that cancer cells experience heightened oxidative stress compared to normal cells, as a consequence of oncogenic transformation, metabolic changes, and increased ROS production [2]. Consequently, the elevated ROS levels in cancer cells can impact redox-sensitive molecules, triggering a cascade of significant outcomes, including cellular proliferation stimulation, modulation of cell differentiation, altered responsiveness to anticancer agents, promotion of genetic mutations and instability, ultimately contributing to carcinogenesis [3]. Therefore, the exploration of novel drugs that regulate mitochondrial dysfunction and trigger the imbalance of reactive oxygen species (ROS) has emerged as a promising avenue for cancer treatment.

Substances derived from herbal medicine play an important role in the development of novel anti-cancer drugs [4,5,6,7,8,9]. Ginseng is used in many cultures, especially in China and other Asian countries, for the treatment and prevention of various diseases, including cancer [10,11]. People who consume ginseng preparations are at lower risk for cancers of the oral cavity, stomach, lung, liver, pancreas, ovaries, and colon [12]. Although only part of the complex mixture of compounds presented in these plants, the ginsenosides, saponin triterpene glycosides, are apparently responsible for most of the pharmacological effects of ginseng and notoginseng. Some ginsenosides assert anti-cancer properties by decreasing DNA synthesis and angiogenesis, reducing host susceptibility to mutation, transformation, and DNA damage, and increasing immunosurveillance and apoptosis [13,14,15,16]. Ginsenosides also enhance the effects of traditional chemotherapeutic agents and protect normal host tissue from damage [10,17,18,19,20].

Methoxylpro-topanaxadiol-3β, 12β, 20-triol (AD-1, Figure 1) and 25-Hydroxypro-topanaxadiol-3β, 12β, 20-triol (AD-2, Figure 1) were isolated from total hydrolyzed saponins collected from the leaves of *Panax notoginseng* and *Panax* ginseng fruits separately [21,22,23,24]. AD-1 and AD-2 can inhibit proliferation of many human cell lines, arrest cell cycle in vitro, and decrease tumor weight without side effects in rodent models. In addition, AD-1 and AD-2 have shown better anti-cancer activity in several cancer cells than commercially marketed ginsenoside-related drugs, such as ginsenoside-Rg3, paclitaxel, and methotrexate [25,26]. Hence, the compounds are ideal anti-tumor candidates. AD-1 and AD-2 are diastereomers with a pair of epimers possessing multiple chiral centers and different C-20 configurations ((20(*R*)-AD-1, 20(*R*)-AD-2, 20(*S*)-AD-1 and 20(*S*)-AD-2), Figure 1) [27]. Some studies report that the difference in stereogenic carbon-20 has a great influence on the pharmacological activity of the isomer in ginsenosides [27]. Therefore, we compared the anti-cancer activity of 20(*R*)-AD-1, 20(*R*)-AD-2, 20(*S*)-AD-1 and 20(*S*)-AD-2 in human cancer cell lines, and investigated the differences in molecular anti-cancer mechanisms between these isomers, which may provide meaningful data for 20(*R*)-AD-1, 20(*R*)-AD-2, 20(*S*)-AD-1 and 20(*S*)-AD-2 to become new anti-cancer drugs.

## 2. Results and Discussion

### 2.1. Effect of 20(R)-AD-1, 20(S)-AD-1, 20(R)-AD-2 and 20(S)-AD-2 on Cell Proliferation

We carried out a side-by-side comparison of anti-cancer activity in various cancer cell lines in vitro. The proliferation of prostate cancer cell lines (C4-2B, PC3, DU145, LnCaP), breast cancer cell lines (MCF-7, Bcap37, MDA-MB-435, MDA-MB-231), a lung cancer line (A549), colon cancer lines (LoVo, HCT-116), a liver cancer line (HepG-2) and a gastric cancer cell line (BGC-823) were examined with four ginsenosides (20(*R*)-AD-1, 20(*S*)-AD-1, 20(*R*)-AD-2 and 20(*S*)-AD-2). The effect of 20(*R*)-AD-1 on cancer cells was better than 20(*S*)-AD-1 and the effect of 20(*R*)-AD-2 was slightly better than 20(*S*)-AD-2. These four compounds showed a strong anti-cancer effect on liver, prostate and gastric cancer cells. 20(*R*)-AD-1 had a strong inhibitory effect on a variety of cancer cell lines. As shown in Table 1, the IC50 values of 20(*R*)-AD-1 and 20(*R*)-AD-2 were almost two times as large as 20(*S*)-AD-1 and 20(*S*)-AD-2, respectively.

### 2.2. Effects of 20(R)-AD-1, 20(S)-AD-1, 20(R)-AD-2 and 20(S)-AD-2 on Apoptosis in BGC-803 Cells

Inducing apoptosis of tumor cells is an important strategy for the treatment of cancer. The caspase family of cysteine proteases has a major role in inducing apoptosis [28]. There are two important apoptosis signaling pathways in the cell, namely the death-receptor apoptosis pathway and the mitochondrial apoptosis pathway, which are initiated by caspase-8 and caspase-9, respectively. The two apoptotic pathways converge on executioner caspases. Among the executioner caspases, caspase-3 plays an important role in apoptosis execution, usually present in the cell in the form of low-activity pro-caspase-3 [10,28], which is degraded to caspase-3 by proteolytic enzymes [29]. Finally, caspase-3 transfers to the nucleus and triggers more than 300 downstream substrates associated with inducing apoptosis [30]. In previous reports, AD-1 and AD-2 were shown obviously induce apoptosis in a variety of tumor cells [21]; however, the differences in apoptosis induced between 20(*R*)-AD-1, 20(*S*)-AD-1, 20(*R*)-AD-2 and 20(*S*)-AD-2 were not investigated.

Therefore, morphological changes were observed to explain the different anti-cancer activities of four compounds in BGC-803 cells. The results demonstrated that the cells treated with 20(*R*)-AD-1 were significantly shrunken. However, the cells treated with 20(*S*)-AD-1 did not show the same shrinkage as 20(*R*)-AD-1. Similar results appeared in cells treated with 20(*R*)-AD-2 and 20(*S*)-AD-2 (Figure 2A). Compared with the control group, the surface of BGC-803 cells showed bubble-like areas when exposed to 20(*R*)-AD-1 and 20(*R*)-AD-2. After staining with DAPI, the treated BGC-823 cells also demonstrated prominent chromatin condensation (Figure 2A).

Annexin V-FITC assay was used to prove that the BGC-803 cell line produced apoptosis. The treatment of BGC-803 cells with 20(*R*)-AD-1, 20(*S*)-AD-1, 20(*R*)-AD-2 and 20(*S*)-AD-2 yielded the cell apoptosis rates of 33.4%, 18.6%, 9.1% and 8%, respectively (Figure 2B). 20(*R*)-AD-1 showed the strongest anti-cancer effects against BGC-823 cells, followed by 20(*S*)-AD-1 and 20(*R*)-AD-2. 20(*S*)-AD-2 showed the weakest anti-cancer effect against BGC-803 cells among the four compounds.

To elucidate the mechanisms, the expressions of several key molecules in pathways were detected. We examined the relative proteins of the apoptosis pathway, including caspase-3, casepase-9 and PARP. We examined the protein levels of Cl-caspase-3, Cl-PARP and Cl-caspase-9. Results of Western blotting showed that the expression of cleaved caspase-3, cleaved caspase-9 and cleaved PARP were increased after treatment with 20(*R*)-AD-1 and 20(*R*)-AD-2 (Figure 2C). However, 20(*S*)-AD-1 and 20(*S*)-AD-2 did not show similar results (Figure 2C). All the data indicated that 20(*R*)-AD-1 and 20(*R*)-AD-2 can effectively induce tumor cell apoptosis compared to 20(*S*)-AD-1 and 20(*S*)-AD-2.

### 2.3. 20(R)-AD-1 and 20(R)-AD-2 Induce Apoptosis through Mitochondrial Pathway

About 2 billion years ago, mitochondria were created by phagocytosis of α-proteobacterium, the precursors of modern eukaryotic cells [31,32]. As an important organelle, mitochondria are present in most eukaryotic cells, make up about 40% of the cytoplasm and have a major role in the regulation of programmed cell death and balance of cellular energy metabolism [31,32]. First, as the main energy-producing cellular organ, mitochondria are particularly important for the entire life process. Second, mitochondria regulate the translocation of pro-apoptotic proteins from the mitochondrial intermediate space to the cytoplasm (i.e., Bax and cytochrome C), then activate a number of cell-death-related signal transduction pathways, finally resulting in programmed cell death (i.e., apoptosis) [6]. The collapse of mitochondrial membrane potential, mitochondrial swelling, and opening of mitochondrial membrane permeability transition pores are the main events of mitochondria due to apoptosis [33].

In order to reveal the molecular mechanisms of 20(*R*)-AD-1 and 20(*R*)-AD-2, mitochondria were observed using transmission electron microscopy in BGC-803 cells treated with the compounds. The results are shown in Figure 3A; the nuclei were irregular in shape, with chromatin concentrated around the nuclear membrane, and some nuclei disappeared after the cells were exposed to 20(*R*)-AD-1 and 20(*R*)-AD-2. Meantime, mitochondria were swollen, mitochondrial cristae were discontinuous, structure disappeared and space widened and vacuolated. Numerous vacuoles were found in the cytoplasm in the 20(*R*)-AD-1 and 20(*R*)-AD-2 groups.

We also examined the MMP and the expression levels of proteins that trigger the mitochondrial apoptosis pathway. Firstly, we investigated the MMP of BGC-803 cells treated with the compounds. The results showed that 20(*R*)-AD-1 and 20(*R*)-AD-2 obviously reduced the MMP in a concentration-dependent manner compared with control, 20(*S*)-AD-1 and 20(*S*)-AD-2 (Figure 4B). The compounds’ effects on the expression levels of Bax and cytochrome C in either cytosol or mitochondria were investigated by Western blot assay. The results suggested that the protein level of Bax was decreased in cytosol, while the expression of Bax was increased in the mitochondria. Meanwhile, cytochrome C was transferred from the mitochondria into cytosol in the 20(*R*)-AD-1 and 20(*R*)-AD-2 groups.

### 2.4. Effects of 20(R)-AD-1, 20(S)-AD-1, 20(R)-AD-2 and 20(S)-AD-2 on Oxidative Phosphorylation and Glycolysis

Otto Warburg demonstrated that tumor cells provide energy through glycolysis to maintain cell growth even in the presence of oxygen [34]. However, some reports propose that the energy supply of tumor cells still relies on mitochondrial oxidative phosphorylation [2,35,36]. To investigate the effect of 20(*R*)-AD-1, 20(*S*)-AD-1, 20(*R*)-AD-2 and 20(*S*)-AD-2 on oxidative phosphorylation and glycolytic in MGC-803 cells. The mitochondrial- and glycolytic-stress assays were carried out using selective substrates/inhibitors of different metabolic states while measuring both OCR and ECAR. The results showed that 20(*R*)-AD-1, 20(*S*)-AD-1, 20(*R*)-AD-2 and 20(*S*)-AD-2 obviously inhibited respiration, basal respiration, coupling efficiency, spare respiratory capacity and glycolysis, glycolytic capacity and glycolytic reserve. Relative to 20(*S*)-AD-1 and 20(*S*)-AD-2, 20(*R*)-AD-1 and 20(*R*)-AD-2 exhibited better inhibition of mitochondrial respiration and glycolysis in BGC-803 cells. Interestingly, 20(*R*)-AD-1 and 20(*R*)-AD-2 effectively inhibited ATP production and exerted almost the same effect as oligomycin, an ATPase inhibitor (Figure 4A,B).

### 2.5. 20(R)-AD-1 and 20(R)-AD-2 Triggered Reactive Oxygen Species Generation, Which Was Involved in 20(R)-AD-1- and 20(R)-AD-2-Induced Apoptosis

ROS are important factors in mediating programmed cell death. Meantime, ROS are also the main targets for the treatment of tumors [37]. Many anti-cancer drugs inhibit the proliferation of tumor cells by up-regulating the production of ROS to induce apoptosis [38,39]. Zhang et al. [40] reported that AD-1 (AD-1), a novel ginsenoside derivative, exhibited an anti-lung cancer property by activating the p38 MAPK pathway and by inducing ROS generation. Based on the above reports, DCFH-DA was employed to examine the level of intracellular reactive oxygen species. The results showed that the level of ROS was remarkably increased after the cells were exposed to 20(*R*)-AD-1 and 20(*R*)-AD-2 in MGC-803 cells. The 20(*R*)-AD-1 and 20(*R*)-AD-2 groups (30 μM) produced much higher ROS than the control group and the 20(*S*)-AD-1 and 20(*S*)-AD-2 groups in MGC-803 cells (Figure 5A).

The relationship between ROS production and the apoptosis induced by 20(*R*)-AD-1 and 20(*R*)-AD-2 was detected in MGC-803 cells. The data showed that apoptosis induced by 20(*R*)-AD-1 and 20(*R*)-AD-2 can be suppressed after cells are pretreated with NAC. The data suggested that 20(*R*)-AD-1- and 20(*R*)-AD-2-induced apoptosis may be strongly related to the production of reactive oxygen species in MGC-803 cells (Figure 5B).

## 3. Materials and Methods

### 3.1. Reagents

20(*R*)-AD-1, 20(*S*)-AD-1, 20(*R*)-AD-2 and 20(*S*)-AD-2 (>98%) were obtained as previously described [27]. MTT reagent was obtained from Sigma Aldrich (St. Louis, MO, USA). DMEM medium, penicillin–streptomycin and FBS (Fetal Bovine Serum) were acquired from Thermo Fisher Scientific Co., Ltd. Dingguo Biotechnology (Shanghai, China) supplied Propidium iodide (PI), DAPI and HRP-conjugated secondary antibodies.

### 3.2. Cell Culture

Prostate cancer cell lines (DU145, C4-2B, PC3 and LNCaP), breast cancer cell lines (MCF-7, Bcap37, MDA-MB-435, MDA-MB-231), lung cancer line (A549), colon cancer lines (LoVo, HCT-116), liver cancer line (HepG-2) and gastric cancer cell lines (BGC-803, BGC-823, SGC-7901, MKN-28) were obtained from the Cell Bank of the Chinese Academy of Sciences (Shanghai, China). Prostate cancer cell lines, breast cancer cell lines, lung cancer line, colon cancer lines and liver cancer line were cultured in DMEM medium. BGC-823 cells were grown in RPMI 1640 medium. DMEM and RPMI 1640 media were provided by American Thermal Sciences Corporation, and 10% (*v*/*v*) fetal bovine serum, 100 mg streptomycin/mL and 100 U penicillin/mL were added. All cells were incubated in humidified air at 37 ℃ and 5% CO_2_.

### 3.3. Cell Viability Assay

In vitro screening of the anti-cancer activity of the 20(*R*)-AD-1, 20(*S*)-AD-1, 20(*R*)-AD-2 and 20(*S*)-AD-2 was performed using the MTT experiment assay. Prostate cancer cell lines, breast cancer cell lines, lung cancer line (A549), colon cancer lines, liver cancer line and gastric cancer cell lines were prepared, inoculated into 96-well plates (5000 cells/well) and cultured overnight. Then, the cells were treated with the different concentrations of compounds (0, 10, 20, 40, 80 μM) for 48 h. Every well was incubated with 10 μL MTT (5 mg/mL) for 4 h, and the culture was terminated and the supernatant discarded. The OD values were measured at 495 nm with Bio-Rad Imark, Berkeley, CA, USA to determine the percentage of viable cells.

### 3.4. Morphological Observation and DAPI Staining

BGC-803 cells were planted in 12-well plates (1 × 10^5^ cells/well) and the different concentrations (30 μM) of compounds or 0.1% DMSO (control) added for 24 h. Then, the plates were observed with an inverted microscope.

BGC-803 cells were planted in 12-well plates (1 × 10^5^ cells/well) and the different concentrations (30 μM) of compounds or 0.1% DMSO (control) added for 24 h. Then, the plates were washed with PBS and 4% paraformaldehyde was added for 30 min. After that, the cells were treated with 100 μM DAPI for 30 min under dark conditions, and a fluorescence microscope was used to observe the cells.

### 3.5. Transmission Electron Microscopy

BGC-803 cells were planted in 100 mm dishes and exposed to the compounds (30 μM) for 12 h; the cells were then collected and fixed overnight with 2.5% glutaraldehyde at 4 °C overnight. The samples were dehydrated by a series of graded alcohols and embedded in resin. Microstructural images of cell samples were captured with transmission electron microscopy.

### 3.6. Flow Cytometry

The early and late apoptotic stages were detected using an Annexin V-FITC Apoptotic Detection Kit (CA). BGC-803 cells were inoculated into 12-well plates (1 × 10^5^ holes/holes) and 30 μM compounds or 0.1% DMSO (control) added for 12 h. PBS reagent was used to collect and wash the BGC-803 cells. The staining of BGC-803 cells utilized PI and Annexin V-FITC, and the apoptosis of BGC-823 cells was detected by flow cytometry.

DCFHDA and JC-1, fluorescent probes, were employed to examine the levels of ROS and MMP, respectively. BGC-803 cells were cultured in 6-well plates (2 × 10^5^ cells/well) and exposed to different concentrations (30 μmol/L) of compounds for 12 h. Subsequently, cells were collected and were incubated with 500 μL DCFHDA and JC-1 (Beyotime Biotechnology, China) (10 μmol/L) for 20 min at 37 °C. Then, the cells were washed with 1 × Incubation Buffer twice. Flow cytometry was employed to examine fluorescence density.

### 3.7. Metabolic Profiling

Seahorse XF glycolysis assay (Agilent Technologies, Berlin, Germany) was employed to examine the mitochondrial respiratory (basal respiration, coupling efficiency and spare respiratory capacity) and the glycolytic function (basal glycolysis, glycolytic capacity and glycolytic reserve) according to the manufacturer’s instructions. BGC-803 cells were incubated in XF8-well cell culture microplates (Agilent) (4 × 10^4^ cells/per well) for 24 h, then were exposed to 20(*R*)-AD-1, 20(*S*)-AD-1, 20(*R*)-AD-2 and 20(*S*)-AD-2 (20 μM) for 4 h. Oxygen consumption rate (OCR) and the extracellular acidification rate (ECAR) were measured for 16 min to establish a baseline measurement. Mitochondrial respiratory and glycolytic functions were examined after the addition of oligomycin (1.0 μM), FCCP (2.5 μM) and a mixture of antimycin-A (0.5 μM), rotenone (0.5 μM), glucose (10 mM), oligomycin (1.0 μM) and 2-deoxy-D-glucose (2-DG) (50 μM), respectively. OCR and ECAR values were investigated after each compound was injected into XF8-well cell culture microplates with a specific time interval.

### 3.8. Western Blot Analysis

Total protein was extracted with RIPA lysis buffer and total protein concentration was determined using the BCA method. Then, the protein samples were separated by SDS-PAGE (10% gel). After SDS-PAGE, the gel was transferred to NC membranes. After blocking with 5% skim milk powder, the membrane was incubated with a 1:1000 dilution of primary antibody against Cl-caspase-3, Cl-caspase-9, PARP, Bcl-2, Bax, cytochrome C and β-actin in 5% BSA and overnight at 4 °C. Of these, Cl-caspase-3, 9 and PARP were purchased from Cell Signaling Technology Company, Danvers, MA, USA. Bcl-2, Bax and cytochrome C were purchased from Proteintech Group, Inc; β-actin was acquired from Ding-Guo Biotech Ltd., Beijing. Then, the membrane was washed with TBST and incubated with a 1:5000 dilution of a horseradish-peroxidase-conjugated secondary antibody for 2 h. Then, the membrane was detected using a chemiluminescence system.

### 3.9. Statistical Analysis

All these data were shown as mean ± SD, and three independent experiments adopted with the SPSS 17.0 software. The data differences were considered significant if *p* < 0.05. Image J 2 and SPSS 17.0 software were employed to analyze the results of Western blotting.

## 4. Conclusions

In conclusion, our findings showed that the anti-cancer activity of 20(*R*)-AD-1 and 20(*R*)-AD-2 is significantly better than 20(*S*)-AD-1 and 20(*S*)-AD-2. 20(*R*)-AD-1 and 20(*R*)-AD-2 induced apoptosis and declined MMP, up-regulated the expression of cytochrome C in cytosol, transferred Bax to the mitochondria, suppressed oxidative phosphorylation and glycolysis and stimulated reactive oxygen species (ROS) production. Apoptosis can be attenuated by the reactive oxygen species scavenger N-acetylcysteine. However, 20(*S*)-AD-1 and 20(*S*)-AD-2 barely exhibited the same results. All results indicated that 20(*R*)-AD-1 and 20(*R*)-AD-2 exhibited better anticancer activity than 20(*S*)-AD-1 and 20(*S*)-AD-2 through suppressing cellular energy metabolism through the mitochondrial apoptosis pathway, which is likely involved in ROS generation. The difference of ginsenoside isomers in anticancer activity was possibly one of the main causes of the spatial selectivity of groups in 20-C. 20(*R*)-AD-1 and 20(*R*)-AD-2 may have had the same spatial conformation, resulting in smaller space steric hindrance; thus, 20(*R*)-AD-1 and 20(*R*)-AD-2 showed better anti-cancer activity.

## Figures and Tables

**Figure 1 molecules-28-06698-f001:**
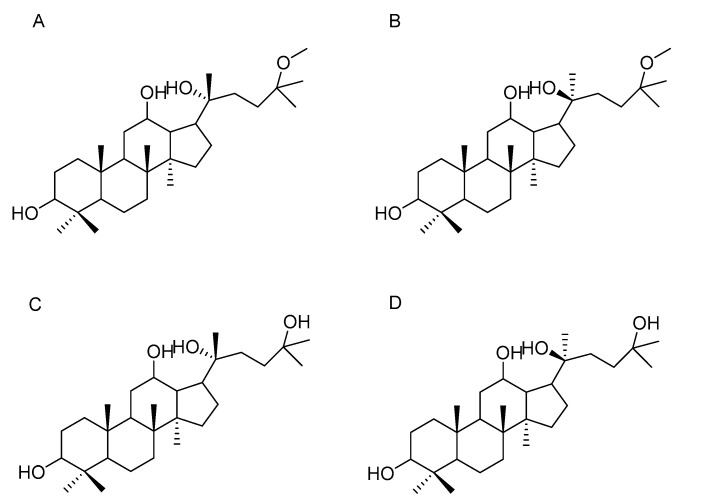
(**A**) Chemical structures of 20(*R*)-25-methoxyl-dammarane-3β, 12β, 20-triol R (20(*R*)-AD-1), (**B**) 20(*R*)-25-methoxyl-dammarane-3β, 12β, 20-triol S (20(*S*)-AD-1), (**C**) 20(*R*)-25-Oxhydryl-dammarane-3β, 12β, 20-triol R (20(*R*)-AD-2), and (**D**) 20(*R*)-25-Oxhydryl-dammarane-3β, 12β, 20-triol S (20(*S*)-AD-2.

**Figure 2 molecules-28-06698-f002:**
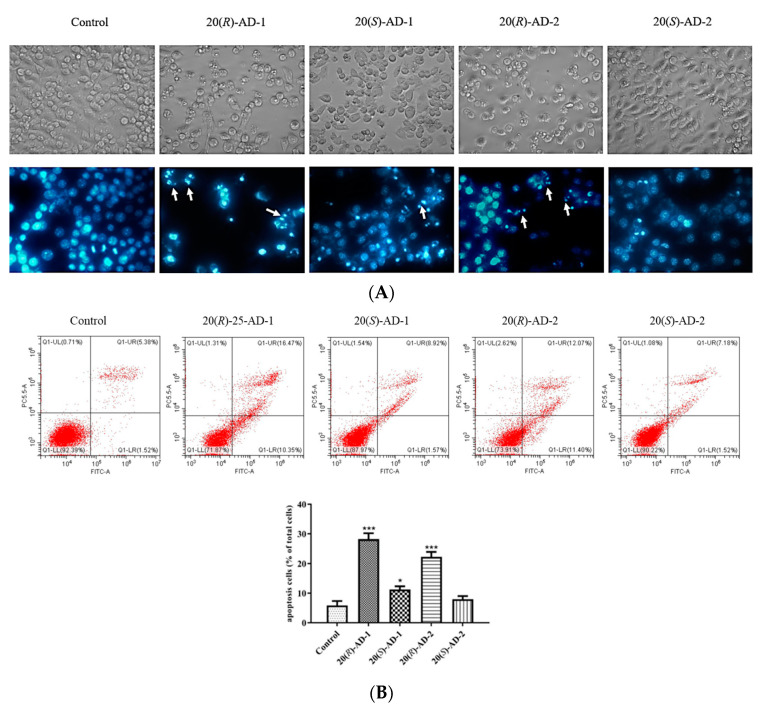
(**A**) Morphological changes induced by 20(*R*)-AD-1, 20(*S*)-AD-1, 20(*R*)-AD-2 and 20(*S*)-AD-2 (30 μM) on BGC-803 cells for 24 h and observed by an inverted microscope. Nuclear morphology of control and treated cells was detected by DAPI staining with a fluorescence microscope; white arrows reveal nuclear condensation. (**B**) Cells were exposed to 20(*R*)-AD-1, 20(*S*)-AD-1, 20(*R*)-AD-2 and 20(*S*)-AD-2 (30 μM) for 24 h, followed by flow-cytometry-based apoptosis assays. (**C**) Cells were exposed to 0, 15 and 30 μM of 20(*R*)-AD-1, 20(*S*)-AD-1, 20(*R*)-AD-2 and 20(*S*)-AD-2 for 12 h, then the apoptosis-related protein levels of Cl-caspase-3, Cl-caspase-9 and Cl-PARP were analyzed by Western blotting. Protein levels were normalized to β-actin. All assays were performed in triplicate. * *p* < 0.05, ** *p* < 0.01, *** *p* < 0.001 compared with the control.

**Figure 3 molecules-28-06698-f003:**
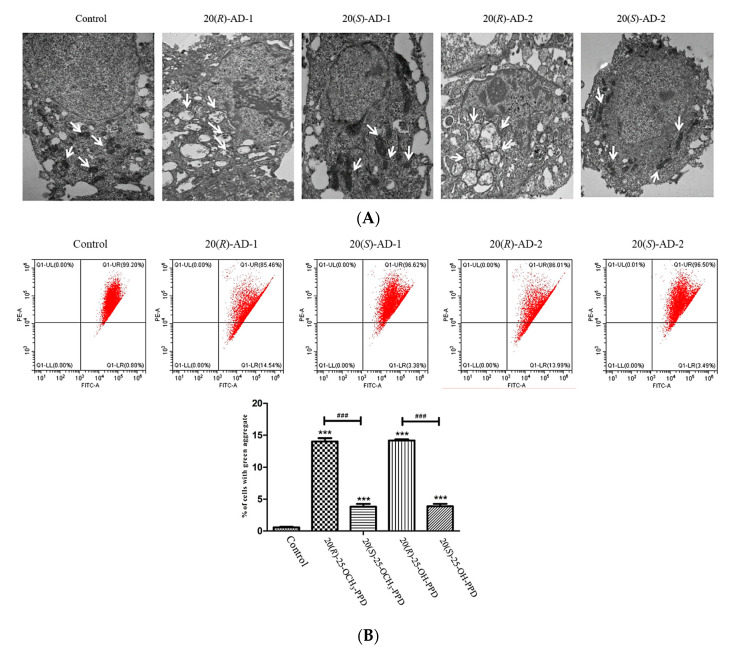
The apoptotic mechanisms of induction by compounds in BGC-803 cells. (**A**) Transmission electron microscopy images of microstructural changes of mitochondria in BGC-803 cells incubated with 20 μM compounds for 12 h. White arrowheads show mitochondria. Scar bar in the original and enlarged images indicates 1.2 μm. (**B**) Cells were treated with compounds and stained with JC-1, and then analyzed by flow cytometry. The relative proportion of red and green fluorescent intensity indicates the mitochondrial depolarization ratio. (**C**) The translocation of Bax and release of mitochondrial cytochrome C were examined by Western blotting assay in BGC-803 cells. Data are expressed as means ± SDs of triplicate experiments performed independently. *** *p* < 0.001 compared with the control, ### *p* < 0.001 20(*R*)-AD-1 compared with 20(*S*)-AD-1 and 20(*R*)-AD-2 compared with 20(*S*)-AD-2.

**Figure 4 molecules-28-06698-f004:**
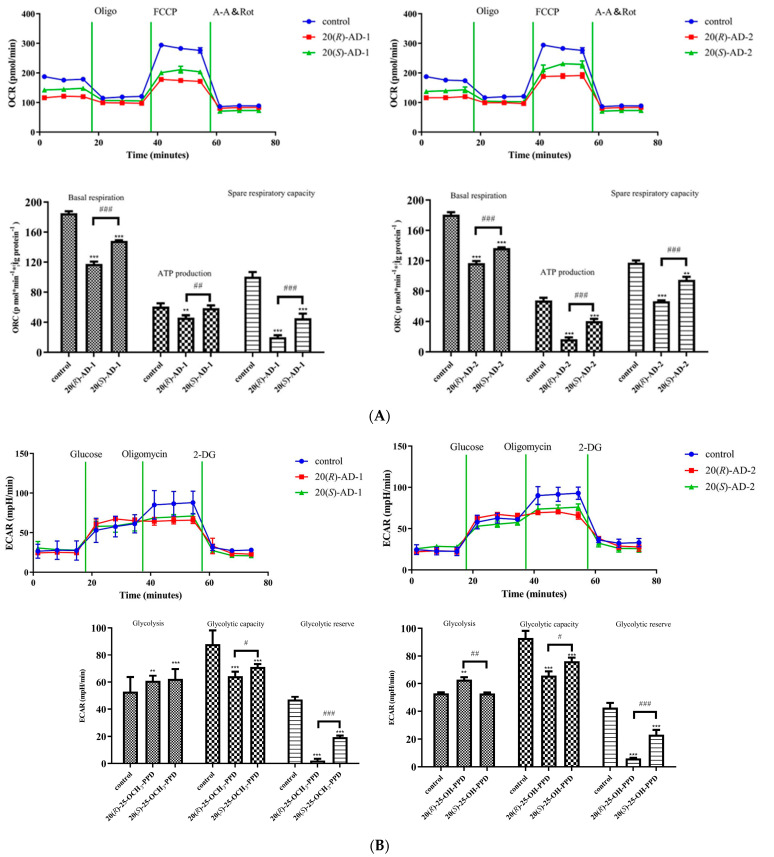
The effects on mitochondrial respiration and glycolysis caused by 20(*R*)-AD-1, 20(*S*)-AD-1, 20(*R*)-AD-2 and 20(*S*)-AD-2 in BGC-803 cells. (**A**) Oxygen consumption rate (OCR) was measured using an XF8 extracellular flux analyzer by adding oligomycin (1 μM), carbonyl cyanide p-trifluoromethoxy-phenylhydrazone (FCCP, 2.5 μM) and rotenone (1 μM)/antimycin A (1 μM) to 20(*R*)-AD-1-, 20(*S*)-AD-1-, 20(*R*)-AD-2- and 20(*S*)-AD-2-treated BGC-803 cells. (**B**) Extracellular acidification rate (ECAR) was measured using an XF8 extracellular flux analyzer by adding glucose (10 mM), oligomycin (1.0 μM) and 2-deoxy-D-glucose (2-DG) (50 μM) to 20(*R*)-AD-1-, 20(*S*)-AD-1-, 20(*R*)-AD-2- and 20(*S*)-AD-2-treated BGC-803 cells. Data are expressed as means ± SDs of triplicate experiments performed independently. ** *p* < 0.01 *** *p* < 0.001 compared with the control, # *p* < 0.05, ## *p* < 0.01, ### *p* < 0.001 20(*R*)-AD-1 compared with 20(*S*)-AD-1 and 20(*R*)-AD-2 compared with 20(*S*)-AD-2.

**Figure 5 molecules-28-06698-f005:**
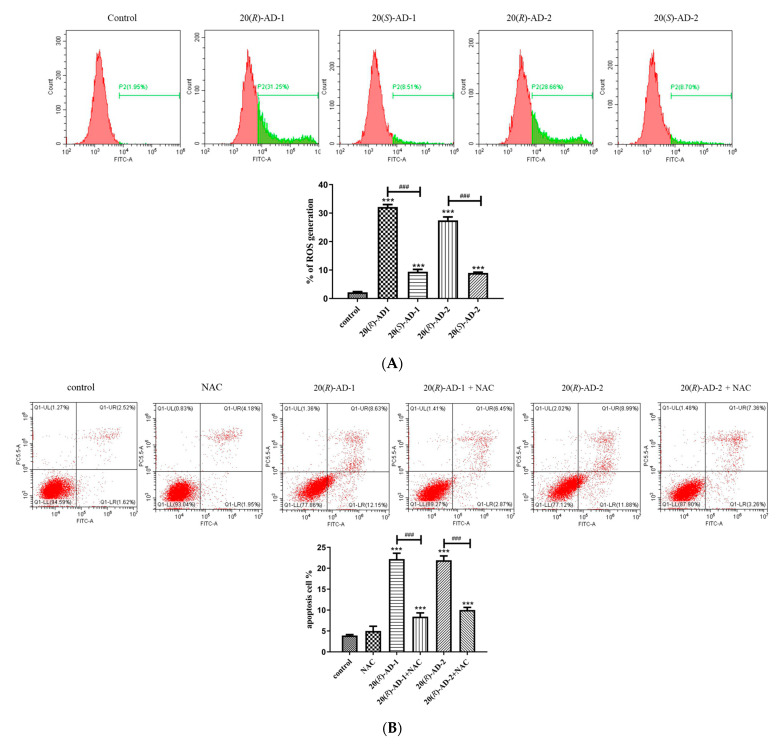
20(*R*)-AD-1 and 20(*R*)-AD-2 induce BGC-803 cell apoptosis involved in reactive oxygen species production. (**A**) Intracellular reactive oxygen species levels were examined by using flow-cytometry-based reactive oxygen species level assays. Representative plots of flow-cytometry-based reactive oxygen species levels are shown. (**B**) Pretreatment with NAC cells and untreated cells were exposed to 20 μM compounds for 12 h, followed by flow-cytometry-based apoptosis assays. Representative plots of flow-cytometry-based apoptosis assays are shown. Group data analysis of the percentage of apoptotic cells is shown. All assays were performed in triplicate. *** *p* ˂ 0.001 compared with the control, ### *p* < 0.001 20(*R*)-AD-1 compared with 20(*R*)-AD-1+NAC and 20(*R*)-AD-2 compared with 20(*R*)-AD-2+NAC.

**Table 1 molecules-28-06698-t001:** The IC_50_ of 20(*R*)-AD-1, 20(*S*)-AD-1, 20(*R*)-AD-2 and 20(*S*)-AD-2 in cancer cell lines.

Cell Type	Cell Line	IC_50_ (μM) 48 h			
		20(*R*)-AD-1	20(*S*)-AD-1	20(*R*)-AD-2	20(*S*)-AD-2
Gastric	BGC-803	8.11 ± 0.36	57.731 ± 1.23	15.519 ± 0.68	55.959 ± 1.26
	BGC-823	12.995 ± 0.62	44.617 ± 0.79	16.120 ± 1.36	61.836 ± 2.18
	SGC-7901	17.109 ± 1.31	51.666 ± 0.94	20.814 ± 1.65	69.522 ± 2.36
	MKN-28	18.558 ± 0.43	45.978 ± 0.69	22.275 ± 0.54	42.583 ± 0.63
Lung	A549	19.618 ± 0.32	46.554 ± 1.38	23.937 ± 0.65	55.341 ± 0.84
Colon	LoVo	21.72 ± 0.12	55.244 ± 0.59	25.713 ± 0.74	54.647 ± 1.63
	HCT-116	20.76 ± 0.53	46.005 ± 1.03	25.099 ± 0.63	49.464 ± 1.54
Liver	HepG-2	19.167 ± 0.18	49.587 ± 0.65	28.179 ± 1.26	59.533 ± 0.87
Prostate	C4-2B	17.11 ± 0.36	38.731 ± 1.23	22.519 ± 0.68	49.959 ± 1.26
	PC3	27.995 ± 0.62	43.667 ± 0.79	35.12 ± 1.36	60.826 ± 2.18
	DU145	39.109 ± 1.31	60.666 ± 0.94	46.814 ± 1.65	88.582 ± 2.36
	LnCaP	20.558 ± 0.43	35.978 ± 0.69	26.275 ± 0.54	41.593 ± 0.63

## Data Availability

https://figshare.com/s/9a14dca37aca00250a1d (accessed on 20 July 2023).

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
