# Peer review of "Stereospecificity of Ginsenoside AD-1 and AD-2 Showed Anticancer Activity via Inducing Mitochondrial Dysfunction and Reactive Oxygen Species Mediate Cell Apoptosis"

_molecules, 2023, doi:10.3390/molecules28186698_

Round 1
Reviewer 1 Report
Dear Authors,
The manuscript can be accepted after addressing the below mentioned corrections.
Comment 1.
The introduction should describe in more detail the relationships between the mitochondrial dysfunction and reactive oxygen species and cancer progression.
Comment 2.
Considering the journal's specifics, please provide the structures of the studied derivatives in Figure 1 using chemical formula editors.
Comment 3.
Please improve the quality of Figures 2-4.
The manuscript needs moderate editing in the English language.
Author Response
Comment 1.
The introduction should describe in more detail the relationships between the mitochondrial dysfunction and reactive oxygen species and cancer progression.
Thanks for the reviewer’s comments. the relationships between the mitochondrial dysfunction and reactive oxygen species and cancer progression have been added in the introduction. the revised portion are marked in red.
Comment 2.
Considering the journal's specifics, please provide the structures of the studied derivatives in Figure 1 using chemical formula editors.
Thanks for the reviewer’s comments. The structures of the studied derivatives have been edited by ChemBioDraw Ulta 14.0
Comment 3.
Please improve the quality of Figures 2-4.
Thanks for the reviewer’s comments. The quality of Figures 2-4 has been improved.
Reviewer 2 Report
In this paper, Wang et. al. studied the anti-cancer activity of Ginsenoside AD-1 and AD-2 and their underlying mechanism. They demonstrated that 20(R)-AD-1 and 20(R)-AD-2 exhibited more strong anti-cancer activity than 20(S)-AD-1 and 20(S)-AD-2 through mitochondria-mediated apoptosis. 20(R)-AD-1 and 20(R)-AD-2 induced apoptosis by increasing ROS generation and concomitant apoptotic processes including the increase of mitochondrial Bax, cytosolic cytochrome c, and caspase-9 activation. Although the authors confiemed anti-cancer activity, they did not propose novel molecular mechanism of apoptosis by inducing 20(R)-AD-1 and 20(R)-AD-2. In light of this and specific comments detailed below, I cannot recommend publication of this work in Molecules.
1. The anti-cancer activity of AD-1 and AD-2 by inducing apoptosis has been reported (ref. 19). Furthermore, AD-1 induced ROS generation was also reported (ref. 39). Therefore, the data for mitochondria-mediated apoptosis such as the increase of mitochondrial Bax, cytosolic cytochrome c, and caspase-9 activation are predictable data. In this regard, the authors did not show sufficient data and novel mechanism for anti-cancer activity of AD-1 and AD-2.
2. Although authors claimed that the differences in apoptosis induced by 20(R)-AD-1, 20(R)-AD-2, 20(S)-AD-1 and 20(S)-AD-2 were investigated (line 22 and 100), they did not suggest the reason why 20(S)-AD-1 and 20(S)-AD-2 exhibited lower anti-cancer activity.
3. Authors showed that AD-1 and AD-2 inhibited mitochondrial respiration, glycolysis, and ATP production in Figure 4. Also, authors claimed that AD-1/AD-2 induced apoptosis was suppressed by NAC in Figure 5. What is the primary target of AD-1 and AD-2 for inducing apoptosis?
Minor editing of English language required.
Author Response
- The anti-cancer activity of AD-1 and AD-2 by inducing apoptosis has been reported (ref. 19). Furthermore, AD-1 induced ROS generation was also reported (ref. 39). Therefore, the data for mitochondria-mediated apoptosis such as the increase of mitochondrial Bax, cytosolic cytochrome c, and caspase-9 activation are predictable data. In this regard, the authors did not show sufficient data and novel mechanism for anti-cancer activity of AD-1 and AD-2.
Thanks for the reviewer’s comments. This article primarily focuses on the investigation of the isomeric anticancer differences between AD-1 and AD-2. Our primary objective is to ascertain whether the isomeric forms of AD-1 and AD-2 exhibit variances in the established mechanisms. Furthermore, we have discovered that the isomeric forms of AD-1 and AD-2 exhibit divergent effects on mitochondrial respiration and glycolysis, revealing novel mechanisms that have not been previously identified.
- Although authors claimed that the differences in apoptosis induced by 20(R)-AD-1, 20(R)-AD-2, 20(S)-AD-1 and 20(S)-AD-2 were investigated (line 22 and 100), they did not suggest the reason why 20(S)-AD-1 and 20(S)-AD-2 exhibited lower anti-cancer activity.
Thanks for the reviewer’s comments. Our experimental findings indicate that 20(S)-AD-1 and 20(S)-AD-2, at the same concentration, are unable to efficiently activate the mitochondrial apoptotic pathway, induce the generation of reactive oxygen species, and inhibit mitochondrial respiration and glycolysis, as compared to 20(R)-AD-1, 20(R)-AD-2. We believe that these differential effects could account for the disparate anticancer activities exhibited by 20(S)-AD-1 and 20(S)-AD-2.
- Authors showed that AD-1 and AD-2 inhibited mitochondrial respiration, glycolysis, and ATP production in Figure 4. Also, authors claimed that AD-1/AD-2 induced apoptosis was suppressed by NAC in Figure 5. What is the primary target of AD-1 and AD-2 for inducing apoptosis?
Thanks for the reviewer’s comments. ROS are constantly generated during cellular metabolism, particularly through the mitochondrial respiration, and exert various physiologic actions. Impaired mitochondrial respiration results in an elevation of reactive oxygen species levels, surpassing the threshold of tumor cells. Consequently, the mitochondrial apoptotic pathway is triggered, ultimately leading to cellular apoptosis. Although we have not identified the precise protein target through which AD-1/AD-2 primarily exerts its effects, our results indicate that AD-1/AD-2 predominantly induces mitochondrial dysfunction, leading to oxidative stress and ultimately initiating apoptosis.
Round 2
Reviewer 2 Report
While the authors have explained about my comments, the manuscript still contains main concerns I raised.
1. The authors claimed that anti-cancer activity of AD-1 and AD-2 by inducing caspase-9 and canspase-3 pathway. The authors should confirm whether caspase-8 is activated. Furthermore, authors should test whether inhibitors against canspase-9 or caspase-3 suppress AD-1/AD-2 induced apoptosis but an inhibitor against caspase-8 does not suppress.
2. Although authors claimed that the differences in apoptosis induced by 20(R)-AD-1, 20(R)-AD-2, 20(S)-AD-1 and 20(S)-AD-2 were investigated, they just showed lower anti-cancer activity of 20(S)-AD-1 and 20(S)-AD-2. Because authors suggest that AD-1/AD/2 induced mitochondrial dysfunction, they should suggest the reason of mitochondrial dysfunction such as mislocalization of mitochondrial membrane proteins.
Minor editing of English language required.
Author Response
- The authors claimed that anti-cancer activity of AD-1 and AD-2 by inducing caspase-9 and canspase-3 pathway. The authors should confirm whether caspase-8 is activated. Furthermore, authors should test whether inhibitors against canspase-9 or caspase-3 suppress AD-1/AD-2 induced apoptosis but an inhibitor against caspase-8 does not suppress.
Thanks for the reviewer’s comments. The death-receptor apoptosis pathway and the mitochondrial apoptosis pathway, which were initiated by caspase-8 and caspase-9, respectively. The two apoptotic pathways converge on executioner caspases. Among the executioner caspases, caspase-3 plays an important role in apoptosis execution, usually present in the cell in the form of low-activity a pro-caspase-3 [1,2], which was is degraded to caspase-3 by proteo-lytic enzymes [3]. Finally, caspase-3 transfers to the nucleus and triggers more than 300 downstream substrates associating with inducing apoptosis [4]. In the preliminary stage, we observed that AD-1 and AD-2 primarily activates the mitochondrial apoptotic pathway involving caspase-9,3. Therefore, we examined the change of caspase-9,3. However, since caspase-8 is part of death-receptor apoptosis pathway, we did not perform any detection for it.
[1] S. Helms, Cancer prevention and therapeutics: Panax ginseng, Alternative medicine review : a journal of clinical therapeutic, 9 (2004) 259-274.
[2] D.R. Green, Apoptotic pathways: paper wraps stone blunts scissors, Cell, 102 (2000) 1-4.
[3] C. Pop, G.S. Salvesen, Human caspases: activation, specificity, and regulation, J Biol Chem, 284 (2009) 21777-21781.
[3] M.O. Hengartner, The biochemistry of apoptosis, Nature, 407 (2000) 770-776.
- Although authors claimed that the differences in apoptosis induced by 20(R)-AD-1, 20(R)-AD-2, 20(S)-AD-1 and 20(S)-AD-2 were investigated, they just showed lower anti-cancer activity of 20(S)-AD-1 and 20(S)-AD-2. Because authors suggest that AD-1/AD/2 induced mitochondrial dysfunction, they should suggest the reason of mitochondrial dysfunction such as mislocalization of mitochondrial membrane proteins.
Thanks for the reviewer’s comments. We also assessed apoptosis induction, mitochondrial dysfunction, as well as mitochondrial respiration and glycolysis in response to 20(S)-AD-1 and 20(S)-AD-2, but the results did not reveal any significant impact on these processes compared to 20(R)-AD-1, 20(R)-AD-2. Based on the experimental findings, we predict that the 20(R)-AD-1 and 20(R)-AD-2 effectively inhibited ATP production and exerted almost the same effect as oligomycin, an ATPase inhibitor. The activity of ATPase is inherently linked to the generation of reactive oxygen species (ROS) [5], thus we posit that by inhibiting ATPase activity, 20(R)-AD-1, 20(R)-AD-2 enhances the production of ROS, consequently inducing apoptosis.
[5] Hana Cho, Yong-Yeon Cho, Min Suk Shim, Joo Young Lee, Hye Suk Lee, Han Chang Kang, Mitochondria-targeted drug delivery in cancers, Biochimica et Biophysica Acta (BBA) - Molecular Basis of Disease, Volume 1866, Issue 8, 2020, 165808, ISSN 0925-4439.